# In Vitro Gastrointestinal Digestion Impact on the Bioaccessibility and Antioxidant Capacity of Bioactive Compounds from Tomato Flours Obtained after Conventional and Ohmic Heating Extraction

**DOI:** 10.3390/foods10030554

**Published:** 2021-03-07

**Authors:** Marta C. Coelho, Tânia B. Ribeiro, Carla Oliveira, Patricia Batista, Pedro Castro, Ana Rita Monforte, António Sebastião Rodrigues, José Teixeira, Manuela Pintado

**Affiliations:** 1CBQF—Centro de Biotecnologia e Química Fina—Laboratório Associado, Escola Superior de Biotecnologia, Universidade Católica Portuguesa, Rua Diogo Botelho 1327, 4169-005 Porto, Portugal; mccoelho@porto.ucp.pt (M.C.C.); tribeiro@porto.ucp.pt (T.B.R.); coliveira@porto.ucp.pt (C.O.); pbatista@porto.ucp.pt (P.B.); pedro_joao_castro@hotmail.com (P.C.); amonforte@porto.ucp.pt (A.R.M.); 2CEB—Centre of Biological Engineering, University of Minho, 4710-057 Braga, Portugal; jateixeira@deb.uminho.pt; 3Association BLC3—Technology and Innovation Campus, Centre Bio R&D Unit, Rua Nossa Senhora da Conceição, 2, Oliveira do Hospital, 3405-155 Lagares, Portugal; 4ToxOmics, CEDOC, NOVA Medical School, NMS, Universidade Nova de Lisboa, 1169-056 Lisboa, Portugal; sebastiao.rodrigues@nms.unl.pt

**Keywords:** *Lycopersicum esculentum*, tomato, byproducts, ohmic heating, bioaccessibility, bioavailability, carotenoids, phenolic compounds

## Abstract

In times of pandemic and when sustainability is in vogue, the use of byproducts, such as fiber-rich tomato byproducts, can be an asset. There are still no studies on the impact of extraction methodologies and the gastrointestinal tract action on bioactive properties. Thus, this study used a solid fraction obtained after the conventional method (SFCONV) and a solid fraction after the ohmic method (SFOH) to analyze the effect of the gastrointestinal tract on bioactive compounds (BC) and bioactivities. Results showed that the SFOH presents higher total fiber than SFCONV samples, 62.47 ± 1.24–59.06 ± 0.67 g/100 g DW, respectively. Both flours present high amounts of resistant protein, representing between 11 and 16% of insoluble dietary fiber. Furthermore, concerning the total and bound phenolic compounds, the related antioxidant activity measured by 2,2′-azino-bis-3-ethylbenzthiazoline-6-sulphonic acid (ABTS) radical cation decolorization assay presented significantly higher values for SFCONV than SFOH samples (*p* < 0.05). The main phenolic compounds identified in the two flours were gallic acid, rutin, and *p*-coumaric acid, and carotenoids were lycopene, phytofluene, and lutein, all known as health promoters. Despite the higher initial values of SFCONV polyphenols and carotenoids, these BCs’ OH flours were more bioaccessible and presented more antioxidant capacity than SFCONV flours, throughout the simulated gastrointestinal tract. These results confirm the potential of ohmic heating to modify the bioaccessibility of tomato BC, enhancing their concentrations and improving their antioxidant capacity.

## 1. Introduction

Tomato pomace is a major byproduct worldwide [1,2] and is well known for its bioactive compounds, e.g., fibers, phenolic compounds [3], and carotenoids, which have a positive impact upon human health, including gastrointestinal health [4,5]. Nevertheless, the health benefits provided by dietary bioactive compounds depends on their bioavailability [5]. According to the Food and Drug Administration, bioavailability “is the rate and extent to which the active ingredient or active moiety is absorbed from a drug product and becomes available at the site of drug action. For drug products that are not intended to be absorbed into the bloodstream, bioavailability may be assessed by scientifically valid measurements intended to reflect the rate and extent to which the active ingredient or active moiety becomes available at the site of drug action”. Components present in food pass through the mouth, stomach, and intestine before reaching the blood [6]. Moreover, bioactive compounds are liberated from food matrices, with modifications within the gastrointestinal tract, solubilization at the intestinal fluids, and permeation through the gut [5,6]. The remaining nonbioaccessible fraction is directed to be used by gut microbiota.

One example is the group of carotenoids that are lipophilic food pigments, including precursors of vitamin A, a nutrient needed for cell differentiation, vision, and immunity [7,8]. Humans cannot synthesize carotenoids, so they are usually consumed with natural sources like fruits and vegetables. In order to exert their function, carotenoids and their metabolites must be assimilated for distribution to tissues and organs [9]. Unlike some other dietary lipids, the bioavailability of carotenoids is impacted by several factors, including food matrix, kind of process, different dietary elements, and nutritional and physiological status [10]. Carotenoids are extremely sensitive to heat and light so they can easily undergo thermal degradation and photodegradation. Therefore, a reliable assessment of carotenoid bioavailability is difficult. The traditional methods used to extract both polyphenols and carotenoids compounds from food matrices require organic solvents, most of them being toxic to human health, and requiring purification methods to be used in the food industry. Currently, several authors propose the use of alternative methodologies to recover these bioactive compounds, such as ohmic heating (OH) [1,11]. 

OH is a process where electrical current passes through a conductor matrix (e.g., food), generating heat through the food’s electrical resistance [11]. Coelho et al. [1] applied this technology to tomato byproducts to extract bioactive compounds. The authors showed that this method could be a selective method to extract both polyphenols and carotenoids using optimized conditions. Furthermore, they only used ethanol–water as a solvent to recover these bioactive compounds, highlighting the potential of this technology to substitute the organic solvents usually used in conventional methodologies.

Although the technology has been successfully applied previously [1], many of the bioactive compounds are not extracted, remaining bound to dietary fibers, which confer biological properties to the solid residue resulting from the extraction [5]. However, the potential antioxidant properties of these solid residues and how their main bioactive compounds are affected by the gastrointestinal tract during digestion were never studied until now. 

Thus, this study aimed to assess the impact of gastrointestinal conditions on bioactives composition and antioxidant activity from tomato flours obtained after two methods of extraction (ohmic heating—OH, and conventional—CONV). 

The bioaccessibility of the major polyphenols and carotenoids were also evaluated, and the cytotoxicity of the digested fractions to guarantee a safe food ingredient with biological properties.

## 2. Materials and Methods

### 2.1. Chemicals

The 2, 20-azo-bis-(2-methylpropionamidine)-dihydrochloride (AAPH), fluorescein, 2, 2-azinobis-3-ethylbenzothiazoline-6-sulfonic acid (ABTS), potassium sorbate, sodium carbonate, ethylenediaminetetraacetic acid (EDTA), sodium sulfite, trifluoroacetic acid (TFA), and sodium lauryl sulfate were purchased from Sigma-Aldrich (Sintra, Portugal). Hexane, acetonitrile, ethyl acetate, Folin–Ciocalteu’s reagent, and potassium persulfate were purchased from Merck (Algés, Portugal). Standards of ascorbic acid, trolox, gallic acid, rutin, *p*-coumaric, and 4-hydroxybenzoic acid, were purchased from Sigma-Aldrich (Sintra, Portugal), while kaempferol, β-carotene, lycopene, zeaxanthin, and lutein (Extrasynthese, Genay, France) were purchased from Extrasynthese (Lyon, France).

### 2.2. Preparation of Tomato Bagasse Flours

Tomato bagasse was kindly provided by Sugal and Italagro group, from the center of Portugal. The two samples were collected and transported immediately to the laboratory. Each sample was characterized for nutritional composition, and a mixture of samples was performed to guarantee a more representative global sample. From there, homogenous samples were immediately dried at 55 °C in a convection oven for 24 h until levels of water activity (a_w_) reached 0.4. The dried solid sample was milled with a kitchen robot (Bimby Vorwerk TM5) and sieved (particle size distribution study using a sieve shaker with a series of sieves, mesh No. 10, 18, 30, 40, 60, 100, and 200) [12,13]. The fractions were combined to obtain tomato bagasse flour (TBF). The TBF represents 86.8% dry weight (DW) of the total solid fraction, and its particle size ranged between 75 to 400 µm. After this, TBF was used to perform the extractions throughout OH using green solvents (70% ethanol, 15 min, 55 °C) and CONV (hexane for carotenoids extraction) [3]. The carotenoids extraction was performed following [14]. Briefly, cold ethanol, 5 mL, was added to 2.5 g of TBF. After the hexane (4 mL) was added and the resulting mixture was centrifuged (10 min at 4000× *g* under 4 °C), and the top layer was removed to a polypropylene tube. The extraction was repeated with 2.5 mL of saturated NaCl and 4 mL hexane. In this process two fractions were obtained, the liquid fraction (LF), which corresponds to the supernatant, and the solid fraction (SF) to the residue. The SF was oven-dried at 55 °C for analysis for 24 h, constituting the solid fraction equivalent to a homogeneous flour. To simplify the designations, the SF of OH tomato byproducts is abbreviated as SFOH, and SF of CONV samples is SFCONV.

### 2.3. Chemical Composition of TBF—Proximate Composition of Tomato Bagasse Flours

Determination of moisture, ash, protein, dietary fiber (TDF), insoluble dietary fiber (IDF), and soluble dietary fiber (SDF) was achieved following the recommendations of the Official Methods of Analysis Chemists [15]. The protein determined by the Kjeldahl method was calculated using a conversion factor of 6.25. All analyses were done in triplicate and expressed as g/kg DW. The moisture content was determined following the method as specified in [15]. The total fatty acids were analyzed according to Coelho et al. [16] with slight modifications. The internal standard used was glyceryl tritridecanoine, TG-C13, and methanol and sodium methoxide were added to 50 mg of sample in the amounts of 2.26 mL and 240 μL, respectively, in the derivatization process. TDF content was estimated using the enzyme–gravimetric method, [17], with some modifications according to [18]. The results were expressed as TDF, insoluble dietary fiber (IDF) and soluble dietary fiber (SDF) g/100 g DW. Free sugar content was determined by Beckman Coulter System Gold HPLC (Knauer, Berlin, Germany) coupled to recover index (RI) and UV detector using Aminex 37-H column (Bio-rad, Berkeley, USA) at 55 °C and 35 mM H_2_SO_4_ as the mobile phase (flow rate: 0.5 mL/min) [13]. The quantification was achieved using standard calibration curves (0.2–2.0 mg/mL) of the following compounds: glucose, xylose, galactose, arabinose, and mannose. All measurements were done in triplicate and expressed as g/100 g DW.

### 2.4. Bioactive Phytochemicals

#### 2.4.1. Phenolic Compound Quantification: Total, Free, and Bound Profiles

Both SFOH and SFCONV were evaluated in terms of free and bound polyphenols to fibers. The SFOH and SFCONV samples obtained after extraction were washed with absolute ethanol three times to extract the soluble free polyphenols (SFPC) and centrifuged for 5 min at 10,000× *g*. Then, for 1 g of washed SFOH and SFCONV, a reaction of 4 h was carried out with 20 mL of NaOH (4 M). The solution obtained was acidified with HCl (6 M) at pH 1.5 to 2.0 and then centrifuged for 30 min at 10,000× *g*. An extraction was then performed with ethyl acetate, for 15 min, five times [19]. The supernatant was concentrated in a vacuum evaporator, resuspended with 10 mL of ethanol, and the polyphenols obtained (bound phenolic compounds, BPC) were analyzed by high-performance liquid chromatography - diode array detector (HPLC-DAD). Results were expressed as mg gallic acid equivalents (GAE)/100 g DW.

The phenolic compounds were released from IDF and SDF fractions using the same hydrolysis process described above. The BPC extract obtained was used to measure the total phenolic compounds (TPC), and to determine the profile of phenolic compounds by LC-ESI-UHR-QqTOF-MS (methodology described in Section 2.4.3). The TPC was evaluated through the Folin–Ciocalteu (TPC) spectrophotometric method, as described elsewhere [1]. To 5 µL of a sample, diluted when needed, 15 µL of Folin–Ciocalteu reagent, 60 μL of sodium carbonate at 75 g/L (Sigma), and 200 μL of distilled water were added, and the solutions were mixed. After samples were heated at 60 °C for 5 min, the OD was read at 700 nm using a spectrophotometric microplate reader (Sunrise Tecan, Grödig, Austria). The content of total polyphenols was expressed as milligram gallic acid equivalent per DW (mg GAE/g). The analyses were performed in triplicate, and a standard deviation was calculated.

#### 2.4.2. Total Antioxidant Activity and Total Carotenoids Content

Total antioxidant activity (AA) was determined using the ABTS radical cation decolorization method as described by Gião et al. [20]. The sample was diluted when needed and added to a colored solution of 2,2′-azinobis-(3-ethylbenzothiazoline-6-sulfonic acid radical cation) (ABTS^•+^). The initial optical density (OD) of the ABTS^•+^ solution, measured at 734 nm using a UVmini 1240 UV–vis spectrophotometer (Shimadzu, Japan), was adjusted to 700 ± 0.020. After 6 min, the final OD was measured and the results given in ascorbic acid equivalent per 100 g DW (ascorbic acid eq./100 g DW).

Oxygen radical absorbance capacity (ORAC) assay was performed according to the method described by Coscueta et al. [21], with some modifications in black polystyrene 96-well microplates (Nunc, Denmark). Antioxidant (20 µL) and fluorescein (120 µL; 70 nM, final concentration in well) solutions were placed in the well of the microplate at 200 µL final volume. A blank (fluorescein + AAPH) using 75 mM phosphate buffer (pH 7.4) instead of the antioxidant solution and eight calibration solutions using trolox (1–8 µM, final concentration in well) as antioxidant were also carried out in each assay. The mixture was preincubated for 10 min at 37 °C. After an AAPH solution (60 µL; 12 mM, final concentration in well) was added. The microplate was immediately placed in the reader and the fluorescence was recorded at intervals of 1 min during 80 min with an excitation wavelength at 485 nm and the emission wavelength at 528 nm. The microplate was automatically shaken before each reading. This assay was performed with a multidetector plate reader (Synergy H1, Vermont, USA) controlled by the Gen5 Biotek software version 3.04. Both AAPH and trolox solutions were prepared daily and fluorescein was diluted from a stock solution (1.17 mM) in the same phosphate buffer. Antioxidant curves (fluorescence versus time) were first normalized to the curve of the blank corresponding to the same assay by multiplying original data by the factor fluorescence blank, t = 0/fluorescencecontrol, t = 0. Results were expressed as µmol TE (trolox equivalent)/100 g DW.

Total carotenoid content was total carotenoids (TC) and assessed using spectrophotometric analysis, as described by Kimura et al. [22], and expressed in milligram equivalent β-carotene per DW (mg β-carotene eq./g DW).

#### 2.4.3. Qualitative and Quantitative Profiles of Polyphenols and Carotenoids

##### HPLC Analysis

Qualitative and quantitative profiles of polyphenols were determined according to the method proposed by [23] with slight modifications. Analysis was conducted on a Waters Liquid Chromatograph (Waters Series 600, Mildford, MA, USA). A C18 guard column (Symmetry^®^ C18) and an Alltech adsorbosil C18 reversed-phase packing column (250 × 4.6 mm i.d. 5 μm particle size and 125 Å pore size) were used for separation throughout this study. The PDA acquisition wavelength was set in the range of 216–600 nm. Mobile Phase: Solvent A with acetonitrile/pure water (5:95 *v/v*), and 0.2% TFA; Solvent B (acetonitrile (100%), pure grade, with 0.2% TFA; flow rate = 1 mL/min. The gradient was 0–2 min (100% B), 2–28 min (60% B), and 28–30 min (100% B).

All the prepared solutions were filtered through 0.45 μm membranes (Fisher Scientific) and the mobile phase degassed before injection onto HPLC. The volume of injection was 20 μL and samples were analyzed in triplicate. Calibration curves were obtained at a detection wavelength 280 nm (gallic acid and 4-hydroxybenzoic acid), 320 nm (*p-*coumaric), 360 nm (rutin). Standard solutions over the concentration range from 0.10 to 100.00 mg/L were prepared for the identification and quantification of the following compounds: 4-hydroxybenzoic acid (y = 10.0 × 10^4^ × − 31507; R^2^ 0.998), *p-*coumaric acid (y = 12.6 × 10^4^ × − 4232; R^2^ = 1); rutin (y = 29.5 × 10^4^ × + 8065; R^2^ 0.999), and gallic acid (y = 48.7 × 10^3^ × − 4520; R^2^ = 0.999) expressed as milligrams per mL of DW. All calibration curves were linear over the concentration ranges tested, with correlation coefficients of 0.999.

Individual carotenoids content was analyzed by HPLC-DAD, in a Vydac 201TP54 C-18 column (250 mm–4.6 mm), equipped with a C-18 reversed-phase packing column (250 × 4.6 mm i.d. 5 μm particle size and 125 Å pore size). The carotenoid profile was performed by chromatographic separation as described by [24]. Solvent A with ethyl acetate (Merck pure grade) and solvent B 90:10 acetonitrile:water (Merck pure grade and pure water, 1.0 mL/min flow rate, at room temperature.

The UV–vis detector was set between 270 and 550 nm. Individual carotenoids were quantified based on a calibration curve built with pure standards: β-carotene, lycopene (y = 2 × 10^8^ x − 44717, R^2^ = 0.999), zeaxanthin (y = 3 x 10^8^ × − 3 x 10^6^ x; R^2^ = 0.999) and lutein (y = 2 x 10^8^ × + 2 x 10^6^, R^2^ = 0.998) and expressed as milligrams per kilogram of DW (mg/Kg DW).

##### UPLC-qTOF MS Analysis

The UPLC-qTOF MS allows an analysis of the complete compound profile and its derivatives, which are not detected by HPLC-DAD. Before analysis, samples were filtered with 0.2 µM PTFE filters and placed in 2 mL vials.

The phenolic and carotenoid analysis was performed on an Ultimate 3000 Dionex UHPLC coupled to an ultrahigh resolution Qq-time of flight (UHR-QqTOF) mass spectrometer (Impact II, Bruker Daltonics, Germany), according to the method described by [13], with slight modifications. Separation of metabolites was performed using a reversed-phase column ACQUITY UPLC^®^ BEH 130Å C18, 1.7 μm, 2.1 × 100 mm (Waters, Milford, MA, USA). For polyphenols, mobile phases were set as follows: A (water 0.1% formic acid); B (acetonitrile 0.1% formic acid); flow, 0.250 mL/min; 0–10 min, 100 to 79% A; 10–14 min, 79 to 73% A; 14–18.3 min, 73 to 42% A; 18.3–24 min, 42 to 0% A; 24–26 min, 0 to 100% A. For carotenoids, mobile phases were set as follow: A (water 0.1% formic acid); B (acetonitrile/methanol (70/30) 0.1% formic acid); flow, 0.250 mL/min; 0–2 min, 15% A; 2–11.6 min, 15 to 0% A; 11.6–14 min, 0 to 15% A; 14–15 min, 15% A [25]. For polyphenols, parameters for MS analysis were set using electrospray ionization (ESI) in negative ionization mode with spectra acquired over a range from *m/z* 20 to 1000 in an Auto MS scan mode. For carotenoids, parameters for MS analysis were set using electrospray ionization (ESI) in positive ionization mode with spectra acquired over a range from *m/z* 50 to 2000 in an Auto MS scan mode [25]. The selected parameters were as follows: capillary voltage, 2.5 kV (negative mode, polyphenols) and 4.5 kV (positive mode, carotenoids); drying gas temperature, 200 °C; drying gas flow, 8.0 L/min; nebulizing gas pressure, 2 bar; collision radio frequency (RF), 300 Vpp; transfer time, 120 μs; and prepulse storage, 8 μs. Postacquisition internal mass calibration used sodium formate clusters, with sodium formate delivered by a syringe pump at the start of each chromatographic analysis. 

The carotenoids data are expressed by the recovery index.

### 2.5. In Vitro Gastrointestinal Digestion

Simulated complete digestion of the two SF samples (SFOH and SFCONV) was performed according to the method described by Ribeiro et al. [5] with slight modifications. This procedure mainly comprised sequential phases simulating different conditions along the gastrointestinal tract. A schematic representation of the main steps performed and the fractions obtained, along with simulated gastrointestinal digestion, are presented in Figure 1, providing information on the process and its conditions. Briefly, 2 g of each sample were mixed with 20 mL of PBS solution, and α-amylase (100 U/mL) was added to the simulated mouth step, and the mixture was incubated for 2 min at 37 °C under agitation (200 rpm). Subsequently, a pepsin (Sigma-Aldrich Chemistry, St. Louis, MO, USA) solution, 25 mg/mL, was added with a ratio of 0.05 mL/mL of a sample at pH 2.0 (simulated gastric solution, stomach) were added following incubation for 2 h, 37 °C at 130 rpm orbital agitation for the gastric phase. Finally, a solution composed of 2.0 g/L pancreatin (Sigma-Aldrich Chemistry, St. Louis, MO, USA) and bile salts, 12 g/L, (Oxoid^TM^, Hampshire, UK) at pH 6.0 was added (0.25 mL/mL) and incubated for 2 h at 37 °C with 45 rpm on an orbital incubator to simulate the small intestinal phase. Then the samples were placed in a dialysis membrane, closed in vials with a solution of PBS to simulate the colon (nonabsorbable sample), and the liquid outside the membrane or blood, basolateral part. Both volumes, inside and outside the membrane, were measured. After the gastrointestinal simulation, the basolateral fraction was freeze-dried for subsequent analysis.

During the gastrointestinal stimulation, samples were collected in each step—mouth, stomach, small intestine, colon, and basolateral fraction—to analyze total phenolic compounds, total carotenoids, individual compounds, and metabolites (phenolic, carotenoids, and volatile compounds) by HPLC and UPLC-qTOF MS, and antioxidant activity. All analyses were performed in triplicate.

### 2.6. Simulated Digestion and Transepithelial Diffusion across Intestinal (Caco-2/HT29-MTX) Cell Layers

Permeability test was assessed in Corning^®^ Transwell embeds, utilizing well plates. Caco-2/HT29-MTX cocultures were seeded into the inserts to imitate absorptive epithelia of the human digestive tract, as detailed beforehand [26]. The human Caco-2 cell line is of a colonic origin, but upon confluence, differentiates to form a well-differentiated polarized monolayer of columnar absorptive cells with a brush border and expressing typical metabolic enzymes and transporters [27].

#### 2.6.1. Cell Culture

Caco-2 (passage 63) and HT29-MTX (passage 55) cells were grown in DMEM supplemented with 10% (*v/v*) fetal bovine serum (FBS), 100 U/mL penicillin and 100 μg/mL streptomycin, and nonessential amino acids in a humidified chamber at 37 °C and 5% CO_2_. The culture media was replaced every 2–3 days until cell seeding.

The cells were seeded on collagen-coated membrane inserts (0.4 μm pore size; Corning, NY, USA) placed in 12-well culture plates. For medium substitution, the medium was removed from the wells and 0.5 and 1.5 mL of new culture medium was added to the apical and basolateral sides, individually. The media was replaced every 2 days for 21 days and the transepithelial electrical resistance (TEER) measured to validate the barrier integrity (TEER > 250 Ω·cm^2^).

#### 2.6.2. Cell Layer Integrity

Transepithelial electrical resistance (TEER) of Caco-2/HT29 cocultures is the measurement of electrical resistance across a cellular monolayer and is a very sensitive and reliable method to confirm the integrity and permeability of the monolayer, using a Millicell^®^ ERS-2 Voltohmmeter (Merck, Germany) [26]. It was performed during the permeability test and estimated along with cell growth and after each example. During porousness tests, TEER values for Caco-2/HT29-MTX were regularly over 250 Ω·cm^2^, indicating that cells retained the integrity of their membranes [26].

#### 2.6.3. Permeability Assay

Upon the arrival of the examination, the culture medium was completely removed. Medium in the basolateral side (receptor part) was supplanted with 1.5 mL of PBS, pH 6.8. After the gastrointestinal simulation, the resultant digested fraction after gut simulation was centrifuge at 2000× *g*, for 5 min, and applied (0.5 mL) into the apical side of transwells seeded with Caco-2/HT29-MTX coculture cell layers. The aliquots were pulled back from the basolateral side at 15, 30, 60, 120, and 180 min. The bioactive compounds released were assessed by LC-ESI-UHR-QqTOF-MS and HPLC, utilizing a similar technique as described above.

### 2.7. Volatile Compounds

Volatile compounds were extracted from SFOH and SFCONV samples using a headspace solid-phase microextraction technique (SPME), at each sampling point from gastrointestinal compartments. The fiber used was a DVB/CAR/PDMS (divinylbenzene/carboxy/polydimethylsiloxane) 50/30 mm from Supelco (Bellefonte, PA, USA). A Varian gas chromatograph, Varian CP-450 (Walnut Creek, CA, USA), equipped with a Varian Saturn 240 MS (Walnut Creek, CA, USA) mass spectral detector was used for the identification and quantification of the volatile compounds. The column used for the volatiles separation was a FactorFour VF-WAXms 15 m length × 0.15 mm of internal diameter × 0.15 µm of film thickness column, from Varian (Lake Forest, CA, USA). Briefly, 1 g of SFOH and SFCONV were dissolved in 5 mL of ethanol solution (10%), in a headspace screw vial, and spiked with 20 μL of 3-octanol (50 mg/L). Samples were preincubated in a CombiPAL oven at 40 °C and 150× *g* for 5 min, and the fiber was exposed after 15 min at 150× *g* for extraction. Desorption of the volatile compounds in the injector was performed at 220 °C for 10 min. All mass spectra were acquired in the electron impact (EI) mode. Compound identification was achieved by comparing retention times and mass spectra obtained from a sample containing pure, authentic standards or by National Institute of Standards and Technology (NIST) database. Compound quantitation was performed based on standard calibration curves using *m/z* quantifiers. 1,2-dimethylindole—equivalents of linalool; 2,6-dimethylbenzaldehyde—equivalents of linalool; benzoic acid—pure standard; β-cyclocitral—pure standard; 3,4-diethenyl-1,6-dimethyl—pure standard; camphenol—pure standard; linalyl acetate—pure standard; linalool—pure standard.

### 2.8. Recovery and Bioaccessibility Indexes of Polyphenolic and Carotenoids Compounds Throughout In Vitro Gastrointestinal Digestion

Bioaccessibility is defined as the percentage of the bioactive compound solubilized after intestinal dialysis step; this index defines the proportion of the bioactive compound that could become available for absorption into the blood system:Bioaccessibility index (%) = (BCS/BCDSI) × 100(1)
where BCS is the bioactive content (mg) in the digested sample after the dialysis step (OUT) and BCDSI is the bioactive content (mg) in the digested sample after the small intestinal step.

The recovery index determines the main compound concentration during each step of gastrointestinal digestion following the equation:Recovery index (%) = (BCDF/BCTM) × 100(2)
where BCDF represents the bioactive content (mg) in each digested fraction and BCTM is the bioactive content (mg) quantified in the test matrix.

### 2.9. Mutagenicity

The reference Ames mutagenic test was used to assess the mutagenicity of the tomato flours after the colon simulated digestive process, using *Salmonella typhimurium* (His-) strains TA-100 and TA-102 following the method described by [16]. The test was performed with and without a metabolic mix in which 0.5 mL of a liver homogenate (S9 mix) was included in the test tube alongside microorganisms and extract. The positive control used was quercetin (Sigma, St. Louis, MO, USA) without metabolic activation (without rat liver extract, e.g., S9 mix addition) and B[a]P (Sigma, St. Louis, MO, USA) with metabolic activation (with S9 mix). The number of revertants was analyzed and compared with the negative control. All the tests were performed in triplicate.

### 2.10. Statistical Analysis

Statistical analysis was done using IBM SPSS Statistics v21.0 (IBM, Chicago, IL, USA). The normality of the data distribution was evaluated using the Shapiro–Wilk test. As the data proved to follow a normal distribution, Student’s t-test and one-way ANOVA, coupled with Tukey’s post hoc test, were used to determine the differences of the mean values between bioactive compounds or bioactivities concentrations along digestion. Supervised cluster analysis (Orthogonal partial least squares discriminant analysis (OPLS-DA)) was applied to evaluate the metabolite patterns of tomato bagasse flour (relative intensity) as a function of time using MetaboAnalyst 3.0 (http://www.metaboanalyst.ca/ (accessed on 3 October 2020)) on log-transformed data after autoscaling (mean-centered and divided by the standard deviation of each variable). The differences were considered significant for *p*-values ≤ 0.05. 

## 3. Results

### 3.1. Characterization of Solid Fractions Obtained after OH and CONV Extraction of Tomato Bagasse

The overall average proximate analysis results for both flours obtained through different extraction methods, SFOH and SFCONV, are presented in Table 1. In dried samples, ash, protein, fat, and fiber were higher for SFOH than SFCONV samples with significant differences for the last three parameters (*p* < 0.05). The higher nutritional value presented by SFOH than SFCONV samples is mainly due to the heating process that occurs during the OH treatment [1], causing pore formation (electropermeabilization), and allowing intracellular component diffusion [28] that probably remained on the SF after LF extraction.

The significantly higher content of protein observed for SFOH than SFCONV could be related to the interference of electric field, frequency, and consequently, temperature caused by OH in the protein aggregation, denaturation, and soluble protein content in the matrix [20,29]. Moreover, the OH on tomato bagasse causes ions and other charged molecules (e.g., proteins) to move toward the opposite charge electrode [30]. Pereira et al. [31] studied the effect of OH on whey protein and showed that fast heating through the joule effect in concomitance with low electric field contributes to obtaining lesser protein aggregates and higher soluble protein content. Regarding the CONV method, this does not use temperature and does not cause denaturation of proteins. Furthermore, this method uses hexane with protein extraction yields similar to ethanol (the latter is used in OH) [32].

Fatty acids were also higher in SFOH than SFCONV samples (*p* < 0.05). OH extraction has been associated with higher oil yield, but also with high quantities of phenolic compounds and higher antioxidant activities due to the more intense breakdown of vegetable tissues caused by heating inside the extraction chamber [33,34]. This higher oil extraction capacity of the OH methodology could justify the higher amount of total fatty acids in SFOH than SFCONV, but the use of water as an extraction solvent in OH methodology is probably the main reason for the higher fatty acids retention in SFOH than SFCONV.

Another nutritional component of tomato flours positively affected by OH extraction was dietary fiber. The SFOH presented higher total fiber and insoluble fiber than SFCONV samples (*p* < 0.05). Ramírez-Jiménez et al. [35] presented similar results with corn flours when comparing the extraction effects of OH and CONV method (nixtamalization, whereby corn is soaked and cooked in an alkaline solution). The results suggest that OH preserves the compounds (e.g., insoluble fiber), not removing the pericarp and aleurone layer during the process, while insoluble fiber losses in the CONV method could be attributed to pericarp removal during this process, which uses organic solvent. The higher amount of insoluble fiber in SFOH makes OH process an important alternative to produce tomato flours with enhanced bioactive profile, preserving bound phenolics fraction commonly associated with insoluble dietary fiber [35].

Although polyphenols are released toward the liquid extracts, phenolics intimately bound to the fiber (but also to protein and lipids) are not extracted and are part of the flour [36]. Thus, not only free phenolic compounds (FPC) were assessed but also bound phenolics compounds (BPC) of SFOH and SFCONV were analyzed to obtain a more complete phenolic characterization from tomato byproducts after phenolics recovery (Table 1). Significant differences (*p* < 0.05) between extracted fractions, SFOH, and SFCONV, were found. SFOH exhibited a higher TPC in FPC than in BPC. Additionally, FPC of SFOH contained a higher amount of TPC than SFCONV (*p* < 0.05). These results could be explained by the extraction solvents used by OH (ethanol–water) and CONV (hexane) extraction. Tomato byproducts are a rich source of liposoluble compounds that are more easily extracted by CONV extraction solvent, which justifies the lower free TPC of SFCONV. The higher TPC richness of free phenolics of SFOH were also reflected in a higher antioxidant activity associated with this flour compared to SFCONV, but only in the ORAC method (*p* < 0.05). These differences probably arise from different mechanisms to measure the antioxidant capacity of each methodology. ABTS is an electron transfer method based on measuring the ability of a potent antioxidant to transfer one electron to reduce radicals, and ORAC measures the ability of an antioxidant to quench free radicals by hydrogen donation (hydrogen atom transfer-based method) [5]. Moreover, ORAC assay is a better model of antioxidant reactions with lipophilic bioactive compounds than ABTS, which can explain the higher ORAC value of the free phenolic fraction of SFOH [37].

Regarding the BPC fraction, no significant differences were detected for both antioxidant methodologies between SFOH and SFCONV (*p* > 0.05), despite the highest TPC value exhibited by BPC of SFCONV compared to SFOH (*p* < 0.05). Despite the significantly lower amount of BPC in SFOH, potent antioxidant compounds remained bound to tomato flour macromolecules (dietary fiber, protein, and lipids). Thus, probably this antioxidant activity was linked to other compounds (e.g., bioactive peptides or lycopene) retained in BPC extract, besides the phenolic compounds but not quantified by Folin–Ciocalteau methodology.

Despite the tomato bagasse fiber richness, until now the fiber profile of SFOH and SFCONV have been neglected. The results of monosaccharides, lignin composition of IDF, and SDF among samples are shown in Table 2. There are differences (*p* < 0.05) in the monosaccharide from SFOH and SFCONV samples. The SDF profile indicates a higher soluble cellulose and mannose rich character for SFCONV than SFOH (*p* < 0.05). Indeed, mannose was not quantifiable in SFOH. The action of OH extraction probably led to the release of polysaccharides rich in glucose and mannose by the cleavage of glycosidic and N-linked bonds, respectively. The release of these neutral sugars from SDF during OH extraction enlightened the lower SDF content of SFOH.

Regarding IDF, SFOH exhibited a higher richness of cellulose (as glucose) than SFCONV. Based on the results, the IDF fraction of tomato flours obtained in this work is essentially a source of cellulose (55.32–50.23 g/100 g fiber) and a less abundant source of hemicellulose (24.72–25.2 g/100 g fiber).

Hemicellulose (as the sum of xylose, galactose, and arabinose) of tomato flours was mainly composed by xylose (13.2–15.1 g/100 g fiber) followed by arabinose (9.31–12.81 g/100 g fiber). The insoluble hemicellulose fraction of tomato flours is essentially a source of arabinoxylans, but SFCONV (9.31 ± 0.51 g/100 g fiber) presented a lower amount of arabinose than SFOH (12.81 ± 0.28 g/100 g fiber). This lower content of arabinose in IDF of SFCONV could be the main reason for its smaller IDF content.

Another significant component of IDF was lignin and resistant protein. In tomato flours, the resistant protein was detected as a significant compound of IDF representing between 11–16% of total IDF. In line with the results obtained above for protein content, SFOH exhibited higher resistant protein content than SFCONV (*p* < 0.05), probably due to the formation of protein aggregates caused by OH [30,31]. The presence of appreciable amounts of resistant protein in IDF has also been previously detected in tomato (≈8% of total IDF composition) [38]. The appreciable amounts of resistant protein of both tomato flours (more significative in SFOH) had the potential to reach the colon, where it can be used by microbiota as a nitrogen source and also as a secondary source of energy, producing ammonia, branched-chain fatty acids, and other metabolites [38]. Finally, Klason lignin were also present in appreciable amounts in IDF of tomato flours (between 13–14% of total IDF composition), in line with the previous studies about tomato fiber composition [39,40]. The lignin, as a complex macromolecule with linked phenolic compounds (BPC), can protect these compounds throughout the gastrointestinal tract up to the colon, where they play a beneficial role in gut health [12].

In general, dietary fiber and polyphenols are usually analyzed separately, however, currently, bound polyphenols are included as components of dietary fiber due to their well-documented association with significant contribution of bound polyphenols to the health-related properties attributed to dietary fiber [41]. The industrial purée process eliminates most of the pulp from tomato fruit, resulting in a byproduct composed mainly of peels and seeds. Therefore, since phenolic compounds are covalently bound to cell wall structural components, such as hemicellulose, cellulose, lignin, pectin, and structural proteins, these samples are rich in phenolics in insoluble forms [42]. In addition, other studies with fruits and vegetables indicate that BPC were largely related to IDF [12,38,43]. It is important to note that BPC can be released and absorbed in the organism during the digestion process [42]; furthermore, they also contribute to antioxidant activity.

As reported before for tomato [38] IDF of tomato flours constitutes a higher source of bound phenolics than SDF (*p* < 0.05). The SFCONV and SFOH exhibited similar amounts of BPC (31.18–34.10 ± 1.21 mg GAE/100 g fiber DW). Concerning SDF, SFOH showed a higher BPC value (7.80 ± 0.43 mg GAE/100 g fiber DW) than SFCONV (4.99 ± 0.35 g GAE/100 g DW), despite the lowest SDF content of SFOH. Comparing the results, OH extraction seemed to be more efficient on the retention of the bound phenolics associated to fiber than CONV extraction. Similar results were found in the literature with corn flours obtained by traditional nixtamalization and ohmic heating process [35].

### 3.2. Total Phenolic Compounds, and Individual Compounds throughout the Digestive Tract and Antioxidant Activity

Although polyphenols are released to the liquid extract, phenolics intimately bound to the fiber are not extracted and remain part of the flour in both SFOH and SFCONV. Thus, bound polyphenols-fiber on both fractions were analyzed to evaluate its potential as antioxidant fiber and its bioaccessibility (Figure 2 and Table 3). The results showed a similar slight increase in total polyphenols content throughout the digestive tract. There is a significant difference between flours (SFOH and SFCONV) (*p* < 0.05) resulting in more OH extraction efficiency than with the CONV method to liquid fraction preparation.

The TPC showed a slight increase in the mouth phase in SFCONV samples (RI 101.42%), while the SFOH decreased significantly (RI 70.44%). A significant increase in the stomach for the SFCONV (RI 124.11%) and SFOH (RI 172.186%) was observed (*p* < 0.05), followed by a decrease in both samples, SFCONV and SFOH, in the intestine (RI 121.99% and 122.18%, respectively).

These increases observed in the stomach could be explained by the compounds presented in both samples, mainly gallic acid, 4-hydroxybenzoic acid, *p*-coumaric acid, and rutin. Furthermore, during this phase, phenolic compounds hydrolysis occurs by enzymatic action and the acidic condition, which releases the phenolics bound to dietary fibers and proteins either in SFCONV or SFOH samples [12,44]. In addition, it is crucial phenomena for the protection of polyphenols against degradation [5]. The significant decrease of TPC observed in SFOH samples (*p* < 0.05) could be explained by the alkaline conditions presented in the small intestine, which promotes the degradation of the compounds.

During the dialysis step, mainly to the basolateral fraction, a significant increase (*p* < 0.05) of TPC was observed in SFOH comparing to the SFCONV samples, probably due to the electropermeabilization (membrane alterations) during the OH process [1,11,44]. Due to this phenomenon, bioactive compounds are better retained and protected within the cell, thus suggesting that they are more available at the time of digestion [44]. Thus, these polyphenols could be released by the additional time interaction (24 h) between samples and both intestinal fluids and digestive enzymes present in pancreatin (a pancreas porcine extract constituted mainly by proteolytic, lipolytic, amylolytic, and nucleic acid splitting enzymes) [5]. Similar results were obtained with enriched apple snacks impregnated with grape juice using the OH method [44].

Taking into account the individual phenolic compounds, the results obtained by HPLC were in accordance with the TPC results. This evidence is clear when compared to the recovery indexes of individual phenolic compounds (Table 3). An example is a *p*-coumaric acid for the SFCONV samples, where a decrease was observed in the mouth (RI 79.35%), and there was an increase in the stomach (RI 121.14%), followed by a decrease in the small intestine (RI 42.49). Another example is rutin, which decreased in the mouth (RI 7.84%) and maintained in the intestine (RI 7.56%). This compound is a water-soluble flavonoid that was quickly absorbed after mouth simulation, which is in agreement with the literature [45,46].

The AA was evaluated by two different methods (ABTS and ORAC). The samples displayed different behaviors. Regarding total AA analyzed by ABTS, no differences were found between SFOH and SFCONV solid fractions (*p* > 0.05); however, for the ORAC results, they presented significant differences between both samples (*p* < 0.05). The different results could be explained by the chemical reaction used, as explained before.

Regarding the AA of SFCONV and SFOH throughout the gastrointestinal tract, different profiles were found for both samples. For SFCONV samples there was a slight increase in the mouth (RI 103.08%), followed by a significant increase in the stomach (RI 134.35%) and a significant decrease in the small intestine (RI 124.14%). For SFOH, a decrease of AA was observed in the mouth (RI 81.45%), followed by a significant increase in the stomach (RI 175.71%) and a significant decrease in the small intestine (RI 108.26%). The results were in accordance with the results reported for TPC and also with the literature [5,44].

Regarding the ORAC method, higher values were found for SFCONV. Nevertheless, the same trend was found for SFOH and SFCONV within the gastrointestinal tract. For the SFCONV sample, a slight increase was observed in the mouth (RI 109.34%), followed by a significant increase in the stomach (*p* < 0.05), and decreasing in the small intestine (RI 108.26%). 

In the case of AA, a reduction was observed in the small intestine step, manifesting more in the ABTS of SFOH samples (RI 63.45%) than ORAC (RI 52.10%). Nevertheless, after the intestinal absorption simulation, the ORAC recovery index is higher in the IN fraction (colon available fraction) than in OUT fraction (basolateral fraction). Despite the initial AA present in both SFCONV and SFOH samples, only a small proportion of the total AA at the end of digestion (IN +OUT) was detected in the basolateral step. The higher content was observed in the basolateral step for SFOH than the SFCONV samples, indicating that the thermal process, caused by OH, significantly affects this property. In addition, better absorption of bioactive compounds from SFOH than from SFCONV was observed.

The digested products’ antioxidant properties depend upon the number of polyphenols and other bioactive compounds available for concentration in the upper or lower parts of the digestive tract (bioaccessibility) or on the quantity absorbed. In addition, the extraction process can cause some physicochemical alterations in flours: on the one hand, the extraction may not cause cleavage of molecules, on the other hand, in the conventional method, using hexane to extract carotenoids and many polyphenols (polar) is not enough to extract these compounds which remain in the flours, probably bound to fiber [47]. During digestion, the molecules present in flours are broken down into smaller ones, and as the bonds between these molecules weaken or break, chemical reactions can occur, and new compounds are produced. These chemical digestion use enzymes through the digestive tract, which break these bonds that hold molecules together, such as polyphenols, carotenoids, fats, proteins, and carbohydrates, which are split into smaller molecules. Polyphenols are commonly bound to the sugar moiety, forming glycones; without the sugar moiety, this simple polyphenol system is the aglycone [48].

These compounds bound to dietary fiber need to be hydrolyzed by enzymes in the gut’s upper region; otherwise, these compounds are not bioaccessible in the intestine but may remain vulnerable to the colonic’s degradation microflora at the large intestine [49]. Given that dietary fiber acts as the entrapping matrix and limits these enzymes’ access to their substrates, most of the polyphenols bound to dietary fiber may pass to the large intestine.

Nevertheless, the smaller molecules pass through the lining of the small intestine and can be absorbed. Thus, they are less bioaccessible than SFOH polyphenols that have polar characteristics and can be incorporated in the polar medium [50,51]. Dietary fibers may interact with polyphenols and carotenoids at the digestive tract, which might improve compound bioaccessibility in the digestive tract and hence their bioactivities. 

This absorption changes the polyphenols extensively; the glycosides are hydrolyzed in the small intestine, or the colon, and release aglycones that can be absorbed. Before entering the bloodstream, the polyphenols undergo different functional changes because of the conjugation activity, primarily in the liver.

Regarding the carotenoids (Table 4), these are hydrophobic compounds and their absorption depends upon effective action from the nutrient array and later solubilization by bile acids and digestive enzymes, culminating in their integration into micelles. The main carotenoid observed in both samples presenting a 527 *m/z* is unknown, with other compounds such as lycopene, phytofluene, and lutein also present. The total carotenoids decreased throughout the gastrointestinal tract (Table 2). They are disassociated from their native surroundings in the tomato byproducts within the extraction process and digested in the stomach (acid conditions and enzymatic hydrolysis). In addition, results showed that OH presents a higher recovery index of lutein and phytofluene than the CONV samples. These significant differences could be explained by the damage that OH does to food structures, which gives access to digestive enzymes and improves bioaccessibility [52,53]. The results showed that lutein and phytofluene from OH (80.74%, 30.18%) and CONV (14.53% and 8.66%) samples presented higher bioaccessibility than lycopene. Nevertheless, the last compound could be cleaved into esters. It appears that these available varieties of fat-soluble vitamins and carotenoids are absorbed by the intestinal mucosa, indicating that these esterified shapes are first hydrolyzed [54,55].

Studies on the subject primarily concern retinyl esters. Their hydrolysis may occur in the stomach, where gastric lipase hydrolyzes about 17.5% of those triacylglycerols. Nevertheless, the information obtained in healthy studies has indicated that gastric lipase does not significantly hydrolyze retinyl palmitate [56]. The hydrolysis of esters of vitamin A, therefore, occurs basically in the duodenum. On the other hand, the individual carotenoids released into the gut lumen are all highly interactive and appear to impede each other’s absorption [57,58,59].

The results observed are promising regarding the general bioavailability improvement of carotenoids and polyphenols, by the expanded penetrability across absorptive epithelia and by the assurance of the bioactive compounds along the gastrointestinal tract.

### 3.3. Carotenoid-Derived Aroma Compounds

Aroma compounds exist in tomato as available volatiles, and they likewise create complexes with nonvolatile precursors such as glycosides, carotenoids, and cinnamic acid derivatives. Carotenoids are essential components of foods and their degradation results in the formation of aroma compounds. Their composition and associated physicochemical attributes are related to significant functions and activities [60]. Thus, we analyzed the volatile compounds produced during the digestion process because degradation of carotenoids leads to a significant production of these compounds (see Table 5). The main volatile compounds detected by GC-MS were 1,2-dimethylindole; 2,6-dimethylbenzaldehyde; benzoic acid; β-cyclocitral; 3,4-diethenyl-1,6-dimethyl; camphenol; linalyl acetate; and linalool. The results presented in Table 5 are according to the literature since the main aroma compounds derived from carotenoids are C13-, C11-, C10-, and C9- derivatives formed via enzymatic reaction of these different carotenoids shown to exist in tomato byproducts flour [61]. As examples, we can identify 2,6-dimethylbenzaldehyde, a C9 produced after alpha-amylase action in the mouth, which, in both samples, in the stomach, was transformed into derivatives by acid hydrolysis. The content of α-cyclocitral—formed by β-carotene-derived apocarotenoid enzymatic action—increased after stomach digestion [61,62]. Nevertheless, while this compound was not absorbed through the conventional method, it was absorbed in the SFOH samples. This result follows the literature, whereas the type of extraction explains the carotenoids’ bioaccessibility differences [10,52,53]. In the conventional method, an organic solvent was used, and many fats were also extracted. In contrast, in ohmic extraction, water–ethanol solution was used and most of the fats remained in the SFOH samples that facilitate the carotenoids’ bioaccessibility regarding SFCONV. Results are in agreement with the described literature, in which samples with a higher amount of fatty acids increase the bioavailability of lipophilic micronutrients, facilitating the solubilization mainly of lipophilic micronutrients in the aqueous phase of digestion [54,56,63,64]. Different mechanisms that occur during the digestion process are responsible for these results. The pancreatic enzymes of gastrointestinal digestion hydrolize on the oil–water interface, making the fatty acids of SFOH more bioavailable. In addition, through small intestine digestion, the carotenoids are incorporated with other lipids, such as cholesterol, phospholipids (present in bile salts), and lipid resultants of digestion products (i.e., free fatty acids, monoacylglycerols, lysophospholipids) into mixed micelles by bile salts. These amphipathic compounds are mainly constituted by water, electrolytes, and organic molecules, including bile acids, cholesterol, phospholipids, and had two essential properties: (1) they are emulsifying agents of lipid aggregates and (2) they solubilize and transport lipids in an aqueous environment [64]. Since biles aggregated into micelles, they can solubilize other lipids in their interstices. The absorption of fat-soluble compounds increases the carotenoids’ absorption capacity due to bile salts. This way, since SFOH has more fatty acids, together with the bile salts they will form more aggregates, mixed micelles in which they will allow greater integration of the carotenes and, therefore, passively pass to the basolateral membrane.

### 3.4. Metabolomics Analysis—LC-ESI-UHR-QqTOF-MS

From the UPLC-qTOF MS analysis, an untargeted analysis was applied based on an established pipeline. In the Venn diagram, 765 features were associated with permeability and only six features related to the treatment (Figure 3).

From the 3D view of the PCA score plot, it was possible to see a clear separation between the mouth samples and the apical zone. Nevertheless, the intracellular and intestine are well clustered together. Besides, with fold changes analysis, a clear separation was observed between flour byproducts after ohmic extraction and conventional extraction, in agreement with reported data. In each sample, differences were also observed throughout the digestion process, which indicates that the flours were metabolized. Furthermore, the OPLS-DA graphical analysis showed that the mouth fraction was more similar to the absorbed fraction in both treatments than with the intestine fraction. According to fold-changes analysis, brassylic acid, didecaniodic acid, and dimethyl-2-oxodecanoylhydroxamic presented significant differences between samples. Thus, the results suggest that the compounds were bioaccessible due to their capacity to be absorbed in the intestine.

### 3.5. Mutagenicity

The mutagenicity of the SFOH and SFCONV samples after the simulated digestion was assessed as some chemical compounds may be harmful or breakdown into mutagenic compounds within the digestive system (Figure 4B).

The results showed an increase in mutagenicity of CONV samples. In contrast, the OH samples were not mutagenic, with no increase in the number of revertants with increasing extract concentrations.

Moreover, the mutagenicity also increased when extracts were exposed to metabolic activation (in contact with S9 mix), suggesting that conventional extract yields compounds with a more significant potential to induce DNA damage when undergoing metabolism. The differences between both extracts may be attributed to less soluble components in the ethanolic extract due to their nonpolar nature. These results are following previous studies [65]. Thus, OH heating is a safer process than the CONV method, which uses organic solvents and could be retained in the flours after extraction, with mutagenic activity.

Regarding the cytotoxicity assessed by the XTT assay, the SFOH extraction presents lower cytotoxicity than SFCONV extracts (Figure 4A). The CONV extraction showed an IC50 of 60 µg/mL. These IC50 values are similar to those measured for medicinal plants [66]. Sharif et al. [65] also obtained mutagenicity and cytotoxicity results of Kalanchoe laciniate n-hexane extracts (corresponding to our conventional method) showing that digested conventional bagasse flour seems to be a source of different toxic compounds against cells. After the digestion procedure, less polar components, such as carotenoids and sterols, are bioavailable as dynamic species. The mechanism of cell toxicity could be due to an arrest of cell cycle at the G0/G1 and G2/M stage and by apoptosis, which has been recently reported for a β-carotene enhanced tomato variety [67]. Some studies have reported impacts of carotenoids, such as neoxanthin and fucoxanthin, on DNA fragmentation in human prostate cancer cells [66,67,68].

### 3.6. Permeability

The Caco-2/HT29-MTX coculture is an alternative method to mimic the human intestinal epithelium. Few studies simulate the absorption of the carotenoids in the epithelia, which could be a good tool to understand the impact of mucins on the bioavailability of bioactive compounds with potential health benefits [69]. Both samples were put in contact with coculture, and the cumulative permeability was studied throughout the simulated digestion to understand the bioavailability of cells and the experienced continuity [26]. In this case, both samples were nontoxic. For both analyses, cell integrity was suitable for analysis (Figure 5). These follow the results presented before, where the SFOH samples present more permeability than SFCONV samples after 60 min. The bioavailability of these compounds depends on the ingredient matrix and the water-soluble metabolites that allows increased permeability [69]. Possibly, the compounds can passively pass the formation of mixed micelles by being more bioavailable.

The results observed are auspicious regarding the overall bioavailability enhancement of both the polyphenols and carotenoids carried, either due to increased permeability across absorptive epithelia or regarding the protection of the bioactive compounds along the gastrointestinal tract.

## 4. Conclusions

In the present work, the chemical composition of two flours obtained by different extraction methods were evaluated. The SFOH presented higher values of IDF fiber and higher protein resistant values than the SFCONV. The study of SFCONV and SFOH throughout a simulated gastrointestinal tract demonstrated that OH increased the bioaccessibility of phenolic compounds and carotenoids. During this process, carotenoids, such as lycopene, lutein, and phytofluene, were identified. Both volatile compounds and metabolic analysis were performed to understand the metabolic pathways of both carotenoids and polyphenolic compounds during the digestion process. The results confirmed that volatile compounds are precursors of carotenoids, and they are absorbed in the bloodstream. Furthermore, the SFOH is safer and contains more bioaccessible bioactive compounds than SFCONV. The information provided in this work gives insight into the impact of alternative methodologies on the bioaccessibility of bioactive compounds from tomato, which allows us to better understand the derived health benefits of tomato bioactive compounds and their potential as a functional ingredient. The introduction of OH technology to recover the bioactive contents could be an opportunity for adding value to the greater byproduct stream from tomato paste production.

## Figures and Tables

**Figure 1 foods-10-00554-f001:**
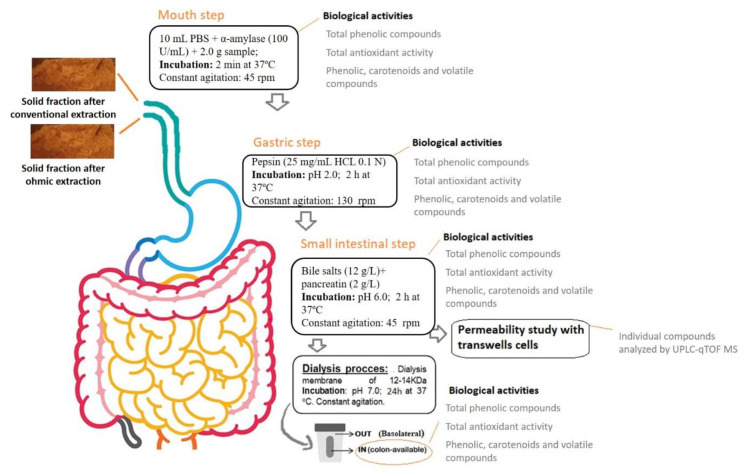
Gastrointestinal simulation and biological analysis throughout each step of digestion.

**Figure 2 foods-10-00554-f002:**
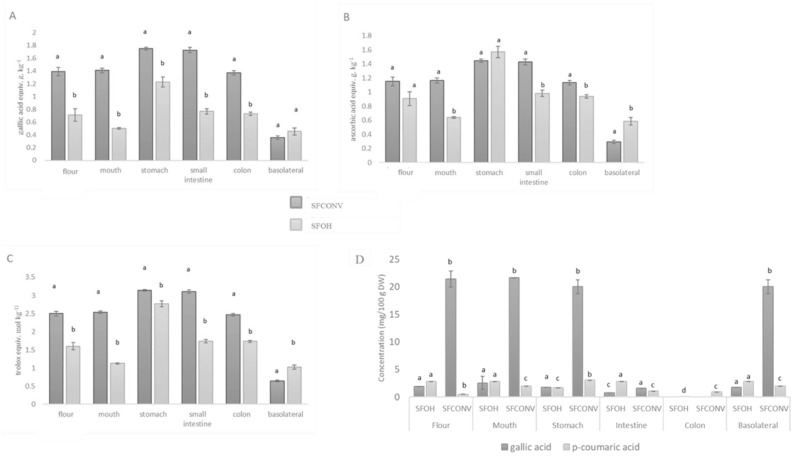
Total phenolics content (**A**), antioxidant capacity by ABTS (**B**), and ORAC (**C**) methods, gallic acid and *p*-coumaric acid concentration (**D**) throughout gastrointestinal tract of samples. SFCONV refers to solid fraction obtained after conventional extraction; SFOH refers to solid fraction obtained after ohmic heating extraction. Significant differences (*p* < 0.05) among extracts for each digested fraction are indicated by Greek letters.

**Figure 3 foods-10-00554-f003:**
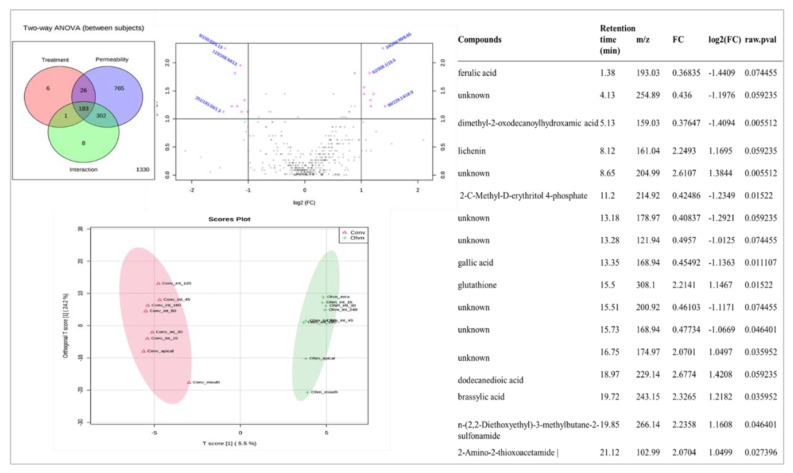
Metabolomic analysis, two-way ANOVA, OPLS-DA, and fold-change analysis.

**Figure 4 foods-10-00554-f004:**
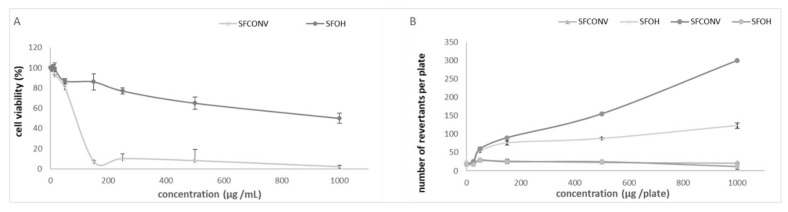
Cytotoxicity (**A**) and mutagenicity (**B**) of SFOH and SFCONV samples. In figure (**B**), 
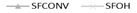
 represent without metabolization and 
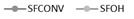
 with metabolization. Results are the means of three determinations ± standard deviation.

**Figure 5 foods-10-00554-f005:**
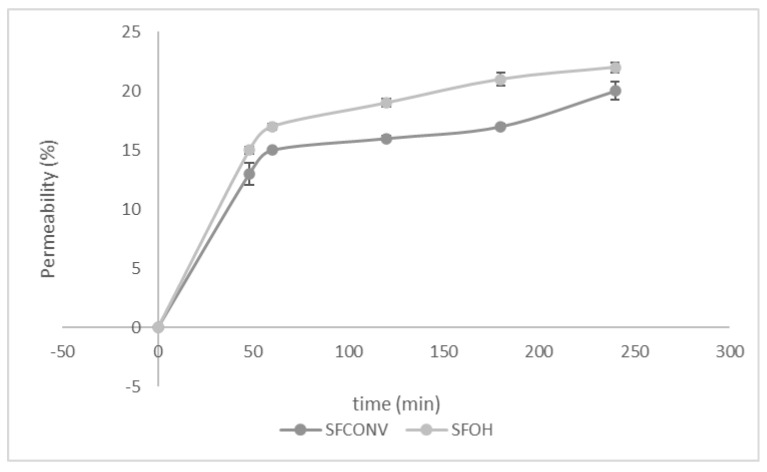
Cumulative permeability of cells to SFOH and SFCONV samples.

**Table 1 foods-10-00554-t001:** The proximate composition, total phenolic compounds, and total antioxidant activity of SFOH and SFCONV (g/100g DW).

Chemical Composition (g/100 g DW)			Sample (g/100 g)	
			SFOH	SFCONV
Proximate composition		Ash	3.32 ± 0.21 ^a^	2.98 ± 0.16 ^a^
		Protein	18.72 ± 0.47 ^a^	16.29 ± 0.59 ^b^
		Total Fatty acids	21.12 ± 0.51 ^a^	17.82 ± 0.3 ^b^
		Dietary Fiber		
		TDF	62.47 ±1.24 ^a^	59.06 ± 0.67 ^b^
		IDF	50.99 ± 0.16 ^a^	46.01 ± 0.13 ^b^
		SDF	10.86 ± 0.85 ^a^	12.98 ± 0.64 ^b^
Phenolic composition	Free phenolic extract (FPC)	TPC	1.46 ± 0.05 ^a^	0.80 ± 0.01 ^b^
		ABTS	0.28 ± 0.01 ^b^	0.99 ± 0.15 ^a^
		ORAC	2.44 ± 0.34 ^a^	1.60 ± 0.09 ^b^
	Bound phenolic extract (BPC)	TPC	0.42 ± 0.03 ^b^	0.73 ± 0.12 ^a^
		ABTS	2.23 ± 0.21 ^a^	2.63 ± 0.11 ^a^
		ORAC	2.41 ± 0.22 ^a^	2.71 ± 0.43 ^a^

Values are the mean of three replicates of three independent experiments ± standard deviation. TDF, total dietary fiber; IDF, insoluble dietary fiber; SDF, soluble dietary fiber; TPC, total phenolic compounds (g gallic acid eq./100 g DW); ABTS, antioxidant activity by ABTS method (g ascorbic acid eq./100 g DW); ORAC, antioxidant activity (g trolox eq./100 g DW). Different letters within the same row are statistically significantly different (determined by ANOVA, *p* < 0.05).

**Table 2 foods-10-00554-t002:** Constituents (g/100 g fiber DW) of SDF and IDF from SFOH and SFCONV samples.

	SFOH		SFCONV	
	SDF	IDF	SDF	IDF
Klason Lignin	n.d.	13.06 ± 0.52 ^b^	n.d.	14.09 ± 0.27 ^a^
Glucose (as cellulose)	24.42 ± 0.43 ^a^	55.32 ± 1.21 ^b^	32.1 ± 0.56 ^c^	50.23 ± 1.76 ^d^
Hemicellulose	15.12 ± 0.54 ^a^	25.2 ± 0.10 ^b^	12.71 ± 1.21 ^a^	24.72 ± 0.30 ^b^
Xylose	*	13.2 ± 0.10 ^a^	*	15.1 ± 0.17 ^b^
Galactose	1.87 ± 0.21	*	*	0.31 ± 0.021
Mannose	*	*	4.10 ± 0.12 ^a^	*
Arabinose	13.25 ± 0.86	12.81 ± 0.28	12.71 ± 1.21	9.31 ± 0.51
Uronic acids^**^	67.41 ± 2.15 ^a^	81.92 ± 1.98 ^b^	58.93 ± 2.56 ^c^	78.49 ± 2.32 ^b^
Resistant Protein	n.d.	16.03 ± 0.05 ^a^	n.d.	11.69 ± 0.03 ^b^
Bond Phenolic compounds ***	7.80 ± 0.43 ^a^	31.18 ± 2.31 ^b^	4.99 ± 0.35 ^c^	34.10 ± 1.21 ^b^

* < LOD; n.d., nondeterminate; IDF, insoluble dietary fiber; SDF, soluble dietary fiber. **, mg GUAE/g fiber DW; ***, mg GAE/100 g fiber DW. Results are the means of three determinations ± standard deviation. Different letters in the same row are significantly different, as determined by ANOVA (*p* < 0.05).

**Table 3 foods-10-00554-t003:** Recovery index and bioaccessibility of total phenolic compounds, antioxidant activity, and individual phenolic compounds from SFCONV and SFOH samples throughout digestion.

Recovery Index (%)							Bioaccessibility (%)
Bioactivities	Samples	Mouth	Stomach	Small Intestine	Colon	Basolateral	
Total Phenol	SFCONV	101.42 ± 2.34 ^a^	124.11 ± 1.28 ^b^	121.99 ± 2.03 ^b^	90.78 ± 1.12 ^d^	9.22 ± 1.07 ^e^	7.03 ± 0.42 ^α^
	SFOH	70.44 ± 1.56 ^a^	172.86 ± 2.34 ^b^	122.18 ± 1.24 ^c^	103.04 ± 1.28 ^d^	15.32 ± 1.25 ^e^	11.14 ± 0.76 ^β^
ABTS	SFCONV	103.08 ± 2.23 ^a^	134.35 ± 1.78 ^b^	124.14 ± 2.02 ^c^	98.49 ± 2.45 ^a^	18.1 ± 1.96 ^d^	14.58 ± 0.13 ^α^
	SFOH	81.45 ± 1.87 ^a^	175.71 ± 2.56 ^b^	108.26 ± 1.95 ^c^	92.51 ± 1.96	7.49 ± 0.06 ^d^	6.47 ± 0.15 ^β^
Orac	SFCONV	109.34 ± 2.25 ^a^	121.65 ± 1.25 ^b^	119.60 ± 1.21 ^b^	90.49 ± 2.54	9.51 ± 0.07 ^d^	7.37 ± 0.31 ^α^
	SFOH	89.19 ± 2.54 ^a^	160.36 ± 2.46 ^b^	108.26 ± 2.38 ^c^	87.50 ± 1.26	12.50 ± 1.01 ^d^	10.35 ± 0.12 ^β^
Phenolic compounds							
gallic acid	SFCONV	101.19 ± 1.87 ^a^	93.56 ± 1.17 ^b^	7.62 ± 0.97 ^c^	n.d.	93.56 ± 1.87 ^b^	92.46 ± 1.03 ^α^
	SFOH	134.31 ± 1.12	8.45 ± 0.2 3 ^b^	29.76 ± 0.79 ^c^	n.d.	100.00 ± 2.05 ^d^	77.07 ± 1.44 ^β^
4-hydroxybenzoic acid	SFCONV	100.38 ± 1.56 ^a^	87.64 ± 1.68 ^b^	12.74 ± 0.26 ^c^	n.d.	87.64 ± 1.67 ^b^	87.31 ± 1.38 ^α^
	SFOH	102.54 ± 1.69 ^a^	84.92 ± 1.78 ^b^	70.98 ± 1.45 ^c^	70.98 ± 0.13 ^c^	13.94 ± 1.36	16.42 ± 1.22 ^β^
*p*-coumaric acid	SFCONV	79.35 ± 2.56 ^a^	121.14 ± 1.27 ^b^	42.49 ± 0.08 ^c^	36.15 ± 0.89 ^d^	78.65 ± 1.23 ^a^	64.92 ± 1.12 ^α^
	SFOH	89.07 ± 1.98 ^a^	59.51 ± 1.25 ^b^	100.42 ± 2.76 ^c^	0.42 ± 0.03 ^d^	99.37 ± 3.01 ^c^	49.74 ± 1.12^β^
rutin	SFCONV	17.01 ± 0.54 ^a^	n.d.	68.83 ± 1.05 ^b^	n.d.	58.83 ± 2.16 ^c^	46.08 ± 1.56^α^
	SFOH	7.84 ± 0.05 ^a^	n.d	n.d.	n.d.	7.56 ± 0.34 ^a^	99.99 ± 0.66 ^β^

Recovery index and bioaccessibility %; n.d., not determined. Results are the means of three determinations ± standard deviation. Different letters in the same row are significantly different (*p* < 0.05), the Greek alphabet symbol means significant differences between methods used in the same column (*p* < 0.05), as determined by ANOVA (*p* < 0.05).

**Table 4 foods-10-00554-t004:** Recovery index and bioaccessibility of carotenoids identified by UPLC-qTOF MS analysis from SFCONV and SFOH samples throughout digestion.

			Mouth	Stomach	Intestine	Colon	Basolateral	Bioaccessibility
Compounds	Mz	Samples	(%)					
n.i.	525	SFOH	84.85 ± 2.31 ^a^	57.97 ± 1.3 ^b^	45.52 ± 1.56 ^c^	40.33 ± 1.87 ^c^	27.17 ± 1.84 ^d^	59.67 ± 1.83 ^β^
		SFCONV	86.32 ± 2.05 ^a^	60.21 ± 1.31 ^b^	49.23 ± 1.55 ^c^	41.32 ± 1.98 ^d^	31.24 ± 2.13 ^e^	63.46 ± 1.76 ^α^
n.i.	527	SFOH	91.55 ± 1.95 ^a^	16.86 ± 0.25 ^b^	15.68 ± 0.99 ^b^	92.48 ± 2.06 ^a^	1.18 ± 1.23 ^c^	7.52 ± 0.74 ^α^
		SFCONV	93.41 ± 1.28 ^a^	89.31 ± 2.45 ^b^	44.66 ± 2.87 ^c^	90.23 ± 2.77 ^a,b^	3.06 ± 0.22 ^d^	6.85 ± 0.32 ^β^
phytofluene	542	SFOH	72.25 ± 1.04 ^a^	36.46 ± 0.78 ^b^	29.14 ± 1.23 ^c^	69.82 ± 1.76 ^a^	8.79 ± 0.97 ^d^	30.18 ± 0.98 ^α^
		SFCONV	26.40 ± 0.27 ^b^	11.53 ± 0.69 ^c^	37.08 ± 1.39 ^a^	35.54 ± 1.43 ^a^	3.21 ± 0.06 ^d^	8.66 ± 0.45 ^β^
lycopene	536	SFOH	n.q	n.q	n.q	n.q	n.q	n.q
		SF	n.q	n.q	n.q	n.q	n.q.	n.q
lutein	569	SFOH	113.9 ± 2.76 ^a^	62.52 ± 1.43 ^b^	57.65 ± 1.43 ^c^	19.26 ± 1.25 ^e^	46.55 ± 1.54 ^d^	80.74 ± 1.32 ^α^
		SFCONV	25.53 ± 0.34 ^c^	11.15 ± 0.98 ^d^	35.85 ± 1.41 ^a^	30.53 ± 1.23 ^b^	5.21 ± 0.09 ^e^	14.53 ± 0.57 ^β^
n.i.	633	SFOH	103.8 ± 2.54 ^a^	58.61 ± 1.77 ^b^	33.06 ± 1.04 ^c^	12.98 ± 0.12 ^d^	35.82 ± 2.31 ^c^	108.32 ± 2.25 ^α^
		SFCONV	n.d	n.d	n.d	n.d	n.d	n.q

Results are the means of three determinations ± standard deviation; n.i., nonidentified; n.d., nondeterminate; n.q., nonquantified. Different letters in the same row are significantly different (*p* < 0.05), the Greek alphabet symbol means significant differences between methods used in the same column (*p* < 0.05), as determined by ANOVA (*p* < 0.05).

**Table 5 foods-10-00554-t005:** Volatile compounds analysis by GCMS.

	Compounds (Name)	1,2-dimethylindole	2,6-dimethylbenzaldehyde	Benzoic Acid	β-cyclocitral	3,4-diethenyl-1,6-dimethyl-	Camphenol	Linalyl Acetate and Linalool
**Samples**	*m/z* quantifiers	133	105	118	119	93	93	93
	Molecular formula	C_10_H_11_N	C_9_H_10_O	C_7_H_6_O_2_	C_10_H_16_O	C_12_H_18_	C_10_H_16_O	C_12_H_20_O_2_/C_10_H_18_O
	MW	145	134	122	152	162	152	196/154
**Conventional method**	CBF	226.6 ± 13.4	0.4 ± 0.1	14.4 ± 4.5	3.3 ± 0.5	0.4 ± 0.1	14.1 ± 0.3	0.0 ± 0.0
	Mouth	190.9 ± 5.4	2.7 ± 0.1	14.0 ± 0.9	2.0 ± 0.1	0.4 ± 0.0	9.2 ± 2.5	0.8 ± 0.3
	Stomach	189.0 ± 3.9	2.2 ± 0.4	4.1 ± 0.8	3.2 ± 0.4	0.9 ±0.1	10.4 ± 1.1	0.3 ± 0.1
	Small Intestine	113.1 ± 4.5	0.0 ± 0.0	1.6 ± 0.6	1.3 ±0.1	0.0 ± 0.0	0.4 ± 0.1	0.0 ± 0.0
	Colon	1.3 ± 0.1	0.0 ± 0.0	0.0 ± 0.0	0.0 ± 0.0	0.1 ± 0.0	1.10 ± 0.3	0.0 ± 0.0
	Basolateral	4.6 ±0.6	0.0 ± 0.0	20.8 ± 3.9	0.0 ± 0.0	0.0 ± 0.0	0.0 ± 0.0	0.0 ± 0.0
**Ohmic conventional**	CBF	232.1 ± 21.7	0.0 ±0.0	8.6 ± 1.4	2.8 ± 0.2	0.1 ± 0.0	2.8 ± 0.3	0.0 ± 0.0
	Mouth	155.5 ± 12.43	1.5 ± 0.3	6.6 ± 1.8	2.6 ± 0.3	0.1 ± 0.0	3.1 ± 0.4	0.3 ± 0.1
	Stomach	208.5 ± 15.84	1.3 ± 0.2	4.6 ± 0.7	4.8 ± 0.5	0.1 ± 0.0	3.7 ± 0.7	0.2 ± 0.0
	Small Intestine	79.5 ± 7.49	0.0 ± 0.0	5.5 ± 0.8	1.3 ± 0.1	0.0 ± 0.0	0.5 ± 0.1	0.0 ± 0.0
	Colon	19.0 ± 4.61	0.0 ± 0.0	2.0 ±0.3	0.0 ± 0.0	0.3 ± 0.1	3.5 ± 0.7	0.0 ± 0.0
	Basolateral	17.3 ± 3.95	0.0 ± 0.0	39.4 ± 4.9	0.6 ± 0.1	0.0 ± 0.0	0.1 ± 0.0	0.0 ± 0.0

Results are given in µg per liter and are a result of the means of three determinations ± standard deviation.

## Data Availability

Data available on request.

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
