# Peer review of "In Vitro Gastrointestinal Digestion Impact on the Bioaccessibility and Antioxidant Capacity of Bioactive Compounds from Tomato Flours Obtained after Conventional and Ohmic Heating Extraction"

_foods, 2021, doi:10.3390/foods10030554_

Round 1

Reviewer 1 Report

Although most of the suggested changes have been made and the work is now better understood, there are still some changes to be made to improve understanding of the results.

General comments:

-Still some subscripts need correction:  aw, chemical structures (H2SO4…).

-It is necessary to add the reference of the commercial companies, country, etc. of important reagents and equipment.

-In the methods that centrifugation is used, change rpm to x g.

-Revise the English language, sometimes the present tense is used instead of the past tense and the singular / plural of the verb does not match the subject.

-As dry weight material is already defined as DW, put the acronym DW whenever this expression is used.

Tables

In general Tables 3 and 4 need to be better edited so that the results can be better seen.

Author Response

Dear Reviewer,

First, the authors sincerely acknowledge the interest demonstrated in our work and the availability to reconsider a revised version of this manuscript. 

We want to thank all your positive inputs and suggestions, which will contribute to improve and enrich this manuscript.

The answers are given just after the transcription of reviewer’ comments, and new information added to the article are highlighted in blue colour as requested in the revised version.

Reviewer comments

Reviewer 1

General comments:

  1. Still some subscripts need correction:  aw, chemical structures (H2SO4…).
  2. The subscripts were improved, please see page 3, line 103; page 3, line 134.
  3. It is necessary to add the reference of the commercial companies, country, etc. of important reagents and equipment.
  4. A paragraph with chemicals, commercial companies and countries was added, please see page 2, lines 86 to 96.
  5. In the methods that centrifugation is used, change rpm to x g.
  6. The changes were performed accordingly, please see page 3, lines 143 and 145; page 7 line 325.
  7. Revise the English language, sometimes the present tense is used instead of the past tense and the singular / plural of the verb does not match the subject.
  8. The English was revised as suggested.
  9. As dry weight material is already defined as DW, put the acronym DW whenever this expression is used.
  10. The sentences were improved accordingly, please see page 3, line.132; page 4, lines 160, 170, 191; page 5, lines 211, 223; page 10, lines 440, 444; table 1; Tables 1 and 2.

Tables

  1. In generalTables 3 and 4 need to be better edited so that the results can be better seen.
  2. The tables were improved accordingly suggestions.

Reviewer 2 Report

In vitro gastrointestinal digestion impact on the bioaccessibility and antioxidant capacity of bioactive compounds from tomato flours obtained after conventional and ohmic heating extraction (foods-1115033)

The Authors provide a very complete and very convincing results about the in vitro digestion, bioaccessibility, cytotoxicity, mutagenicity and antioxidant activity of tomato flour extracts obtained after conventional and ohmic heating extraction processes.

I recommend this work to minor revisions:

  • Some spaces are missing (e.g., g/100gDW line 21; samples(p<0.05) line 24, and after)
  • Delete “ (line 44)
  • Line 180 mg β-carotene eq./g DW. Add DW
  • HPLC: precise injection volume
  • Provide standard curves equations and R2 for each compound (line 200)
  • Dry product (line 210). Use DW instead to be homogenic.
  • Table 2 replace * symbol since its meaning is different from one table to another (LOD or statistical significance).
  • Table 3 is difficult to read. Please change its orientation.
  • Same remark for Table 4.
  • Figure 4 legend is difficult to read too.

Author Response

Dear Reviewer,

First, the authors sincerely acknowledge the interest demonstrated in our work and the availability to reconsider a revised version of this manuscript. 

We want to thank all your positive inputs and suggestions, which will contribute to improve and enrich this manuscript.

The answers are given just after the transcription of reviewer’ comments, and new information added to the article are highlighted in blue colour as requested in the revised version.

Reviewer comments

Reviewer 2

In vitro gastrointestinal digestion impact on the bioaccessibility and antioxidant capacity of bioactive compounds from tomato flours obtained after conventional and ohmic heating extraction (foods-1115033)

The Authors provide a very complete and very convincing results about the in vitro digestion, bioaccessibility, cytotoxicity, mutagenicity and antioxidant activity of tomato flour extracts obtained after conventional and ohmic heating extraction processes.

I recommend this work to minor revisions:

  1. Some spaces are missing (e.g., g/100gDW line 21; samples(p<0.05) line 24, and after). R. The spaces were added accordingly sugestions. Please see page 1, lines 21 and 24; page 3, line 132; page 4 line 170; page 10, lines 443, 448, 457, 458, 460, 462, 463.
  2. Delete “ (line 44). R. The “” in page 1, lines 39 to 44, refers to the bioavailability mean according by Food and drug Administration.
  3. Line 180 mg β-carotene eq./g DW. Add DW. R. The acronym was added accordingly suggestion, please see page 4, line 191.
  4. HPLC: precise injection volume. R. The injection volume 20 ul was added to HPLC methodology, please se page 5, line 205.
  5. Provide standard curves equations and R2 for each compound (line 200). R. The standard curves and R2 were added. Please see page 5 lines 209 and 210.
  6. Dry product (line 210). Use DW instead to be homogenic. R. The changes were performed accordingly.
  7. Table 2 replace * symbol since its meaning is different from one table to another (LOD or statistical significance). R. The changes were performed accordingly.
  8. Table 3 is difficult to read. Please change its orientation. R. The table orientation was changed as sugeested.
  9. Same remark for Table 4. R. The change was performed accordingly.
  10. Figure 4 legend is difficult to read too. R.The legend was imoroved accordingly to suggested.

This manuscript is a resubmission of an earlier submission. The following is a list of the peer review reports and author responses from that submission.

Round 1

Reviewer 1 Report

The present research work titled “In vitro gastrointestinal digestion impact on the bioaccessibil-ity and antioxidant capacity of bioactive compounds from to-mato flours obtained after conventional and ohmic heating ex-traction” is an interesting an innovative work. The research carried out is well focused and the results are appropriate discussed.

General comments:

Please, revise the general considerations for manuscript preparation.

In the first place, authors must use the Microsoft Word template  to prepare their manuscript. Using the template file will substantially shorten the time to complete copy-editing and publication of accepted manuscripts. Continuos number lines is recommended to facilitate the revision task as it figures in the template.

The following sections need to be incorporated: Graphical Abstract,  Author Contributions.

In the text, reference numbers should be placed in square brackets [ ]. Some references need to include the brackets, please revise.

Correct subscripts: aw, chemical structures (H2SO4…), Molecular formula in Table 5.

Remove abbreviations from the text that are not used, example LF, IDF-BPC, SDF-BPC…

Specific comments:

Materials and Methods

2.1. Preparation of tomato bagasse flours

Add more data about companies where the samples were acquired: country…

Regarding sampling: were the samples from both companies mixed?

What is the meaning of “a pooling of samples was performed”? please explain better how the sampling was performed.

Why were the extractions of the compounds were carried out employing different solvents? OH using 70% ethanol, (15 min, 55 ºC) and CONV using hexane. In the latter case, the extraction conditions are not described. Nor do they appear in the referenced bibliography [2]. In this sense, in this study, the obtained results not only are affected by the different techniques OH and CONV, but also are affected by the different solvent used.

“The LF was stored in the freezer at -80 ºC for future analysis”: what analysis were performed in LF in this study?

2.2.2. Analysis of sugars. Sugars should be considered into proximate composition in paragraph 2.2.1.

2.3 Bioactive phytochemicals

“The SFOH and SFCONV samples obtained after extraction were washed during the extraction process to extract the soluble free polyphenols”: what did the washing process consist of?

2.3.2. Determination of total dietary fibre (TDF) composition: This section should be included in the section 2.2.1.

TPC and AOX acronyms are not described in this section.

2.4. In vitro gastrointestinal digestion

The description does not match the information in the figure 1: check time at each stage, add enzyme concentrations in the text and correct step times…

Acronyms used in in figure 1 must be described.

2.6. Phenolic compounds and carotenoid analysis

This section should go within the section 2.3. Total Antioxidant Activity should be described separately.

Regarding Total antioxidant activity (ABTS method) why is it expressed in ascorbic acid equivalent? why is it not expressed in Trolox equivalent as ORAC analysis?

PEITC acronyms is not described in this section.

2.6.2.1–. HPLC analysis

Why in this case results are expressed as micrograms per mL of fresh weight biomass of tomato? why is it not expressed per dry weight material as in the rest of the work?

particle size and pore size of the column used for total carotenoids content analysis?

2.8. Recovery and bioaccessibility indexes of polyphenolic and carotenoids compounds throughout in vitro gastrointestinal digestion

BCDFE: explain better how was obtained. What is “– end of digestion”?.

“BCDF represents the bioactive content (mg) in the digested”, What is in the digested”?

Revise when you put BCDFE or BCDF, it is confusing.

RESULTS:

3.1. Characterization of Solid Fractions obtained after OH and CONV extraction of tomato bagasse.

“Furthermore, this method (CONV method) uses methanol with protein extraction yields similar to ethanol (the latter is used in OH)”. was methanol used? wasn't hexane in CONV method?

Fatty acids analysis is not include in Materials and Methods section

ABTS and ORAC results were expressed per 100g DW, it does not match with material and method information.

How do you know that antioxidant activity is only produced by phenolic compounds? other bioactive compounds may be affecting: carotenoids, vitamin C ...

Tables and figures

In general table edition needs to be improved.

Table 1. antioxidant activity should be included in the title.

“the asterisk within the same column are statistically significantly different”: Change column by row.

Table 2. “Different letters in the same column are significantly different”: Change column by row.

If "* means <LOD", what do the gaps in the table 2 mean? Not determined? non identified? Then add n.d or n.i as in Table 3 and 4.

Figure 4: Please edit figure 4 better. In Cytotoxicity figure A, SFOH does not appear. the Y axis name overlaps the scale.

Reviewer 2 Report

This is an average quality scientific report, but maybe interested to readers if it can be improved. The volatile study was not convincing at all, the identification was too rough, no standard compounds were used, the quantitation using SPME does not meet the standard of flavor analysis, the discussion on volatiles may not be fully accuracy. In addition, the inclusion of Mutagenicity does not make sense, it is too preliminary!

Reviewer 3 Report

The manuscript represent interesting data and on my opinion it falls within the scope of Foods. The presentation and discussion of the results is very comprehensive and some data are of particular interest for the industry. Conclusions highlight the main achievements of the work. However, manuscript should be improved. Below I put my comments and suggestions:

Abstract

In my opinion, the abstract must be rewritten. Please revise the first sentence it does not make sense. In addition, please explain the meaning of SFCONV, SFOH and CONV. Full name of abbreviation is required when it is used first time.

Materials and methods

Materials and methods section is well described and the analytical procedures were correctly applied. However, there are some points that should be revised prior to publication.

Some methods need more description, especially HPLC analysis of polyphenols.

Page 6 - Please provide detailed HPLC method. Which mobile phase do you use, flow rate, gradient, temperature?

Page 7 - Please provide detailed UPLC-qTOF-MS analysis. Which mobile phase do you use, flow rate, gradient, temperature, software used to instrument control, data acquisition and evaluation e.t.c?

Page 7, section 2.7 - The first sentences “Volatile compounds derived from carotenoids are widely distributed in the world, and they are precursors of some powerful aromas in nutrients, e.g., fruits [33]. They ex-hibit high aromatic potential and are thus of great benefit to the manufactures of aromas and fragrances” should not be included in the methodology. Please move this sentence to a more appropriate section.

Results and Discussion

The discussion of the results is very detailed and sometimes the authors repeat the same concepts so the paper it´s heavy to read.

On page 9, the authors state that tomato by-product are a rich source of liposoluble phenolic compounds. Please give examples of such polyphenolic compounds and refer to the relevant references.

References

References did not follow the style of this journal. Authors have to check and revise these errors carefully.

Authors should unify the font throughout the manuscript (including tables).